# Reduced injection risk behavior with co-located hepatitis C treatment at a syringe service program: The accessible care model

**Claire So Jeong Lee[1]\*, Pedro Mateu-Gelabert[2], Yesenia Aponte Melendez[2], Chunki Fong[2], Shashi N. Kapadia[3], Melinda Smith[3], Kristen M. Marks[3], Benjamin Eckhardt[4]**

**1** University of Toronto, Department of Medicine, Toronto, Canada, **2** City University of New York Graduate School of Public Health and Health Policy, New York, New York, United States of America, **3** Weill Cornell Medicine, New York, New York, United States of America, **4** New York University School of Medicine, New York, New York, United States of America

\* claire.lee@medportal.ca

## Abstract

### Background

The main mode of transmission of Hepatitis C in North America is through injection drug use. Availability of accessible care for people who inject drugs is crucial for achieving hepatitis C elimination.

### Objective

The objective of this analysis is to compare the changes in injection drug use frequency and high-risk injection behaviors in participants who were randomized to accessible hepatitis c care versus usual hepatitis c care.

### Methods

Participants who were hepatitis C virus RNA positive and had injected drugs in the last 90 days were enrolled and randomized 1:1 to an on-site, low threshold accessible care arm or a standard, referral-based usual care arm. Participants attended follow-up appointments at 3, 6, 9, and 12 months during which they answered questions regarding injection drug use frequency, behaviors, and treatment for opioid use disorder.

### Primary outcomes

The primary outcomes of this secondary analysis are the changes in the frequency of injection drug use, high-risk injection behaviors, and receiving medication for opioid use disorder in the last 30 days.

### Results

A total of 165 participants were enrolled in the study, with 82 participants in the accessible care arm and 83 participants in the usual care arm. Participants in the accessible care arm were found to have a statistically significant higher likelihood of reporting a lower range of

**Data Availability Statement:** The data from our study are not publicly available due to ethical and legal restrictions as outlined in our IRB approval

and consent forms. The data includes potentially sensitive information about participants, such as personal health details, demographic information, and details regarding illegal behavior, which could compromise participant confidentiality and privacy. According to the Cornell Medicine Institutional Review Board (IRB) approval and consent forms, data that could potentially identify participants will not be shared publicly to ensure compliance with ethical standards and protect participant confidentiality. We remain committed to transparency and will provide access to aggregated data or specific data points that do not compromise participant confidentiality upon reasonable request. Data requests may be sent to Michele Kiely Dr.P.H at Michele.Kiely@sph.cuny.edu.

**Funding:** PMG R01 DA041298. National Institute on Drug Abuse. https://urldefense.com/v3/__https://nida.nih.gov/__;!!MXfaZl3l!b70Uz3LVJ43HL6f5o4ZErrq5DeJ1N31Vw_1CNkCQkosDKV9Do1lpMY2ja0zuxq1ZvMeTCFPtnOh_yCbXFZk6usQ$. No SNK K01 DA048172. National Institute on Drug Abuse. https://urldefense.com/v3/__https://nida.nih.gov/__;!!MXfaZl3l!b70Uz3LVJ43HL6f5o4ZErrq5DeJ1N31Vw_1CNkCQkosDKV9Do1lpMY2ja0zuxq1ZvMeTCFPtnOh_yCbXFZk6usQ$. No.

**Competing interests:** I have read the journal's policy and the authors of this manuscript have the following competing interests: B. E., S. N. K., and K. M. M. have received research grants to their respective institutions from Gilead Sciences Inc.

injection days (accessible care-by-time effect OR = 0.78, 95% CI = 0.62–0.98) and injection events (accessible care-by-time effect OR = 0.70, 95% CI = 0.56–0.88) in the last 30 days at a follow-up interview relative to those in the usual care arm. There were no statistically significant differences in the rates of decrease in receptive sharing of injection equipment or in the percentage of participants receiving treatment for opioid use disorders in the two arms.

## Conclusion

Hepatitis C treatment through an accessible care model resulted in statistically higher rates of decrease in injection drug use frequency in people who inject drugs.

## Introduction

Hepatitis C virus (HCV) infections is a major domestic and international public health concern as the complications of chronic infection, most notably liver cirrhosis, are associated with significant morbidity and mortality [1]. In response to the public health burden of HCV, the World Health Organization (WHO) has announced their goal for the elimination of viral hepatitis by year 2030 [2]. Although the recent introduction of the highly effective direct-acting antiviral (DAA) agents has planted a new-found optimism for reaching this goal, as of 2020 there was an estimated 66,700 new HCV infections in the US with over 2 million people living with chronic HCV infections [2–6]. Evidently, despite the advent of DAA agents, there is a lag in the advancement of effective prevention and accessible health care.

Intravenous drug use is one of the top risk factors for HCV infections, and the incidence of infection has increased by 294% within people who inject drugs (PWID) in the US between 2010 and 2015 [7]. As of 2020, targets needed to meet the Centers for Disease Control and Prevention's (CDC) goal of reducing rates of infection among PWID by 2025 have not been met [8]. Sharing of injection equipment such as syringes and cookers are some of the high-risk injection behaviors for HCV transmission [7]. Once infected, psychosocial and logistical barriers impede PWID from accessing prompt diagnosis and treatment. Some of these psychosocial challenges include patients' fear of stigmatization and apprehension towards healthcare institutions, competing priorities (ex. housing, food, finances, etc.), and co-morbid substance use disorders and psychiatric illnesses [9, 10]. Additionally, stigma in medical settings towards people who use drugs and concerns of continued high-risk injection behaviors associated with re-infection reduce physicians' enthusiasm for offering treatment [11]. These concerns have also motivated policies which hinder PWID from accessing DAA agents, such as some payers requiring sobriety or drug counseling before covering DAA agents [12–15]. Despite the fact that concerns of high-risk behaviors and re-infection heavily influence access to care, there is a paucity of information on behavioral change once PWID receive treatment. The little data we have is promising as a systematic review from 2019 found that the majority of studies included saw an association between treatment and reduced odds of injection use in the past month during treatment and at different time intervals post-treatment [16]. There was no significant decrease in needle and syringe sharing associated with treatment, and data looking at sharing of other injection equipment was conflicting [16]. As for harm reduction through medications for opioid use disorders (MOUD), the current literature highlights low retention rates in treatment programs [17]. Our study aims to investigate as to whether the associated decrease in injection use is amplified in patients who receive accessible care (AC) compared to standard

care, and as to whether the different care models have an impact on high-risk injection behaviors and treatment with MOUD.

The expansion of low-threshold models for HCV screening and treatment of PWID is necessary to reach the goal of HCV elimination [18]. We previously reported results of a randomized controlled trial of the Accessible Care (AC) model which compared a supportive and on-site low-threshold treatment model (AC) to a referral and patient navigation-based model (usual care). [9, 19, 20]. The AC arm demonstrated significantly higher sustained virologic response (SVR) rates compared to the UC arm from higher rates of advancement along the care cascade in the AC arm. Successful treatment itself, in addition to a supportive care model, can be motivating for healthier lifestyles and choices [9, 20].

To assess the changes in injection behaviors and practices of PWID associated with HCV treatment we analyzed data from the Accessible Care study [6]. The focus of this paper is to assess and compare the changes in injection behaviors, including injection frequency and frequency of high-risk injection behaviors, between PWID who received the AC intervention and usual care (UC) intervention.

## Methods

### Study participants

Potential participants were recruited at various sites around New York City and enrolled at the Lower East Side Harm Reduction Center (LESHRC). The LESHRC is a community based, non-profit organization, also acting as a syringe service program (SSP), focused on harm reduction and reducing transmission of HIV/AIDS and HCV amongst PWID in the Lower East Side. Individuals unaware of their HCV status were offered free testing for HCV antibody and polymerase chain reaction (PCR) testing. Eligible participants were above the age of 18 years, spoke English or Spanish, had injected illicit drugs for a minimum of 1 year, had injected at least 1 illicit drug within the last 90 days, and were HCV RNA positive within the last 90 days. Those who were pregnant, diagnosed with advanced liver disease (defined as decompensated cirrhosis and/or hepatocellular carcinoma), and those already receiving HCV care (defined as having attended 2 or more visits with a HCV clinician in the last 6 months) were excluded from the study.

### Study design

Eligible patients with positive HCV RNA tests were randomized 1:1 to the AC or UC arms. All participants provided written informed consent, and the study was conducted in accordance with Good Clinical Practice and the ethical principles that originated in the Declaration of Helsinki. The study was approved by the Weill Cornell Medicine and NYU School of Medicine institutional review boards.

**Accessible care.** The AC model represented a low-threshold model for HCV treatment while providing a supportive harm reduction framework. Participants were connected to an on-site treatment team comprising of a study physician and study care coordinator. The on-site team facilitated a non-stigmatizing atmosphere by maintaining a friendly and informal relationship with the participants and not pressuring for the adoption or rejection of any specific behaviors. Flexible appointments, drop-ins, on-site phlebotomy, and proactive outreach for missed visits were available. Participants not enrolled in or ineligible for insurance were helped to obtain insurance or connected to pharmaceutical drug assistance programs. Patients were able to choose their own dispensing schedule for medications. The care coordinator provided support regarding treatment navigation, insurance, and adherence. Exclusive to the AC arm, all participants receiving DAA therapy were offered an on-site, interactive re-infection

prevention oral presentation adapted from the validated Staying Safe Intervention program [21].

**Usual care.** Participants in the UC arm were connected to an on-site HCV patient navigator who was funded through the New York City Department of Health and part of the Check Hep C program. The patient navigator referred participants to HCV specialists at external hospitals/clinics for treatment. Similar to the AC arm's care coordinator, the patient navigator also provided social support and assistance with insurance navigation. Contrastingly, support for treatment initiation and completion was not provided to the UC group. There was no on-site education regarding re-infection prevention offered [23].

**Assessment.** Participants were followed for a total of 12 months post-enrollment. Each participant underwent a structured baseline interview prior to randomization and subsequent interviews at 3-, 6-, 9-, and 12-months. These interviews were private, lasted between 60 to 120 minutes, and included questions about substance use, health care utilization, overdose, drug injection practices and risk behavior, drug treatment programs, SSP utilization, and hepatitis C treatment. Interviews were conducted in-person or over the telephone, with all interviews prior to the COVID-19 outbreak in March 2020 occurring in-person at the SSP. Participants were compensated equally in both arms for partaking in the quarterly research visits through 12 months at $50 to $70 per visit. Participants were not compensated for engaging in clinical care or receiving HCV treatment; however, AC and UC participants were compensated $40 for undergoing HCV PCR testing 12 weeks after completing treatment (for those who initiated treatment) and 12 months after enrollment (for all participants).

**Study outcomes.** This paper is a secondary analysis of the data from the Accessible Care study. The primary outcomes of this analysis were changes in the frequency of high-risk injection behaviors (sharing of injection equipment and use of previously used syringes), injection frequency (number of days of injection and total number of injection events in the last 30 days), and uptake of MOUD at 3, 6, 9, and 12 months relative to baseline. Data regarding the study outcomes was self-reported via an interview that patients participated in at each timepoint.

## Statistical analysis

The baseline characteristics of participants were reported as frequencies with percentages and mean values with standard deviations. Comparisons between the study arms were tested with of chi-square, Fisher exact, t-test, or Wilcoxon ranked sum tests when appropriate.

Injection frequencies and frequency of equipment sharing were categorized into ordinal variables, rather than continuous variables, to mitigate the impact of extreme outliers, address the highly skewed variable distribution, and to distinguish between key outcomes such as no injection days and daily injections (30 injections days).

At baseline (Table 2) and at each study timepoint (Table 3), participants were grouped into ordinal categories of a range of number of injection events/days in the last 30 days based on their interview answers. The ranges for the ordinal categories were chosen to proportionately capture the distribution of answers from all participants while highlighting no injection days and 30 injection days. The categories for injection days in the last 30 days are as follows: 10 days (no injection), 1–8 days (2 or less times a week), 9–29 days, and 30 days (daily injection). The categories for injection events in the last 30 days were the following: 0 times (no injection), 1–29 times (on average, less than once a day), 30–89 times, and 90+ times (on average, more than 3 times daily). The same ordinal categories were used for equipment sharing events in the last 30 days.

To assess the intervention (AC versus UC) effects observed during the study period (baseline, 3-, 6-, 9-, 12-months), we modeled the data using generalized linear mixed model (GLMM) to evaluate the main effect of intervention, time, and the interaction effect of intervention and time. Additionally, initiation to treatment and its interaction with intervention were included as covariates, accounting for potential confounding effect among those who initiated treatment vs those who did not. Given our focus on testing for trends, time was treated as a numeric variable. For the treatment of opioid use disorder, since the variable is dichotomous, a binary distribution with logit link function was used. For injection frequency and equipment sharing frequency variables, a multinomial distribution with cumulative logit link function was used. All models included a random intercept.

The missingness of the data was also assessed via logistic regression for the five outcome variables at 12 months, employing the outcomes at the previous four data points as predictors [22]. This analysis aimed to examine if the patterns of missingness could be explained by the observed values. In addition, regardless of whether the missingness depended on the observed values, we conducted a sensitivity analysis using multiple imputation.

Statistically significant results were evaluated at p<0.05. Analyses were conducted using SPSS 25 (IBM Corp., 2017) and GLMM was performed using PROC GLIMMIX and multiple imputation using PROC MI/MIANALYZE in SAS software (Release 9.4 SAS Institute Inc., Cary, NC USA).

## Results

Between July 2017 and March 2020, 572 individuals were screened for eligibility. The most common reasons for exclusion were negative/no HCV RNA testing and no injection drug use within the last 90 days (Fig 1). A total of 165 participants were enrolled with 82 participants assigned to the AC arm and 83 to the UC arm. Study participants' baseline characteristics and behaviors pertaining to injection drug use are presented in Table 1. The majority of

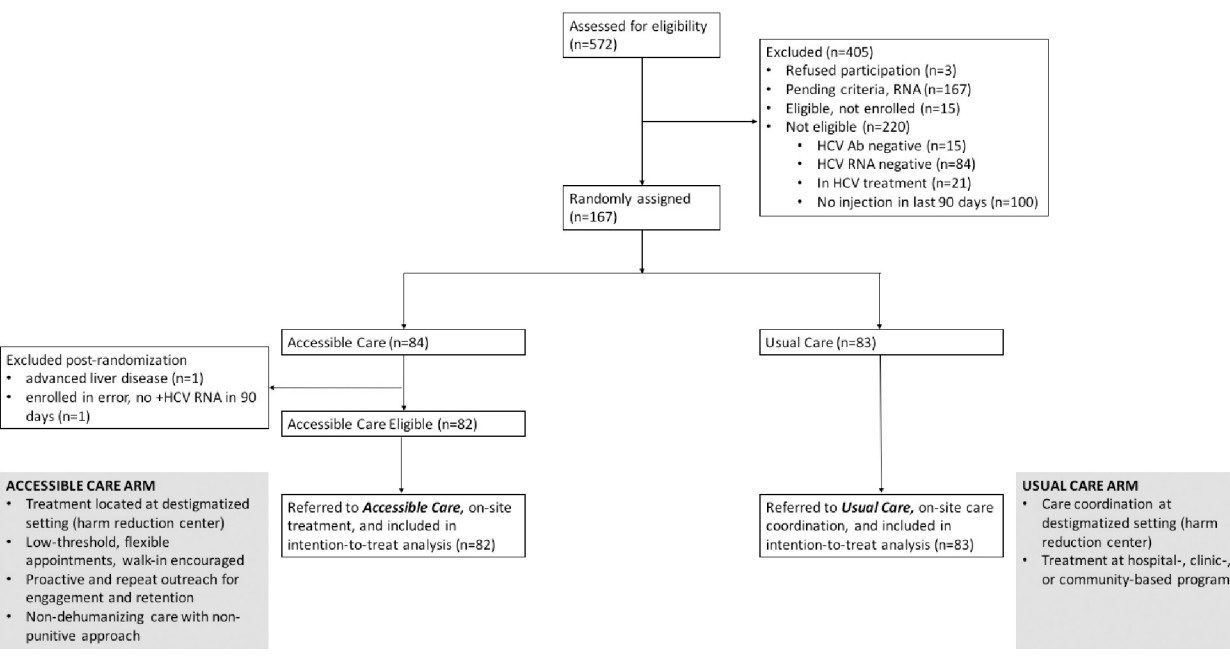

**Fig 1. Consort diagram of accessible care study participants.**

**Table 1. Baseline participant demographics.**

| Characteristic | | Overall (N = 165) n (%) | Accessible Care (N = 82) n (%) | Usual Care (N = 83) n (%) |
|---|---|---|---|---|
| Age *(Mean, sd)* | | 42.0 *(10.6)* | 42.6 *(10.7)* | 41.3 *(10.6)* |
| Age | 18–29 years | 21 (12.7) | 11 (13.4) | 10 (12.0) |
| | 30–44 years | 77 (46.7) | 34 (41.5) | 43 (51.8) |
| | 45+ years | 67 (40.6) | 37 (45.1) | 30 (36.1) |
| Gender | | | | |
| | Male | 128 (77.6) | 62 (75.6) | 66 (79.5) |
| | Female | 36 (21.8) | 19 (23.2) | 17 (20.5) |
| | Transgender | 1 (0.6) | 1 (1.2) | 0 (0.0) |
| Race/Ethnicity | | | | |
| | Hispanic | 97 (58.8) | 45 (54.9) | 52 (62.7) |
| | Non-Hispanic White | 53 (32.1) | 26 (31.7) | 27 (32.5) |
| | Non-Hispanic Black | 8 (4.8) | 7 (8.5) | 1 (1.2) |
| | Non-Hispanic Other | 7 (4.2) | 4 (4.9) | 3 (3.6) |
| Drugs used regularly (last 90 days) | | | | |
| | Heroin | 79 (47.9) | 36 (43.9) | 43 (51.8) |
| | Cocaine | 46 (27.9) | 24 (29.3) | 22 (26.5) |
| | Speedball | 41 (24.8) | 20 (24.4) | 21 (25.3) |
| | Other[a] | 39 (23.6) | 19 (23.2) | 20 (24.1) |
| Homeless (past 3 month) | | 94 (57.3) | 48 (58.5) | 46 (56.1) |
| Health Insurance | | | | |
| | Public | 155 (93.9) | 76 (92.7) | 79 (95.2) |
| | Other | 5 (3.0) | 1 (1.2) | 4 (4.8) |
| | None | 5 (3.0) | 5 (6.1) | 0 (0.0) |
| Incarceration History | | | | |
| | Ever Incarcerated | 138 (83.6) | 70 (85.4) | 68 (81.9) |
| | Recent Incarceration (within the last 90 days)[b] | 12 (7.3) | 2 (2.4) | 10 (12.0) |
| Interaction with non-HCV clinician the past 90 days | | 82 (49.7) | 44 (53.7) | 38 (45.8) |

[a] Other drugs included crack, methamphetamine, prescription opioids, benzodiazepines, and fentanyl

[b] Statistical difference between AC and UC with p<0.05

participants in both treatment arms were male (77.6%), with a mean age of 42.0. The AC group had more participants aged 45 or older than the UC group (45.1% vs. 36.1), while the UC group had more participants aged 31–44 (51.8% vs. 41.5%). There were slightly more Hispanic participants in the UC arm (62.7%) than in the AC arm (54.9%). The UC arm had 51.8% of participants report the use of heroin in the last 90 days compared to the 43.9% in the AC arm. In both groups, more than half of study participants experienced homelessness in the last 3 months. Most participants (84%) had a history of incarceration with 7.3% having been incarcerated within the last 90 days. Most participants (93.9%) were publicly insured. More participants in the AC arm had seen a non-HCV clinician in the last 90 days (53.7%) than participants in the UC arm (45.8%). The baseline characteristics of the two arms were balanced except for the percentage of participants in the UC arm incarcerated in the last 90 days (12.0%) being significantly higher than in the AC arm (2.4%).

At the beginning of the study, 39.4% of study participants were using injection drugs daily with the average number of days of use being 15.5 in the last month (Table 2). In the last 30 days, the sharing of cookers and syringes occurring at an average of 10.1 and 2.4 times

**Table 2. Baseline injection behaviors.**

| Characteristic | | Overall (N = 165) | Accessible Care (N = 82) | Usual Care (N = 83) |
|---|---|---|---|---|
| | | n (%) | n (%) | n (%) |
| **Injection frequency in the past 30 days** | | | | |
| Daily | | 65 (39.4) | 33 (40.2) | 32 (38.6) |
| Less than daily | | 100 (60.6) | 49 (59.8) | 51 (61.4) |
| **No. of days of injection drug use the past 30 days** | | | | |
| Mean (std) | | 15.5 (13.0) | 16.3 (12.9) | 14.7 (13.2) |
| Categories | | | | |
| 0 | | 26 (15.8) | 13 (15.9) | 13 (15.7) |
| 1–8 | | 45 (27.3) | 17 (20.7) | 26 (31.3) |
| 9–29 | | 31 (18.8) | 19 (23.2) | 12 (14.5) |
| 30 (daily) | | 65 (39.4) | 33 (40.2) | 32 (38.6) |
| **No. of injection events the past 30 days** | | | | |
| Mean (std) | | 71.1 (88.4) | 83.6 (92.6) | 58.4 (82.6) |
| Categories | | | | |
| 0 | | 26 (15.8) | 13 (15.9) | 13 (15.7) |
| 1–29 (less than once a day) | | 48 (29.1) | 17 (20.7) | 31 (37.3) |
| 30–89 | | 38 (23.0) | 20 (24.4) | 18 (21.7) |
| 90+ (3 or more times daily) | | 53 (32.1) | 32 (39.0) | 21 (25.3) |
| **No. of sharing cookers the past 30 days** \| Mean *(sd)* | | 10.1 *(30.6)* | 6.9 *(20.9)* | 13.3 *(37.7)* |
| **No. of used syringe used by some else the past 30 days** \| Mean *(sd)* | | 2.4 *(11.8)* | 3.3 *(14.4)* | 1.4 *(8.7)* |
| **Current Medication for Opioid Use Disorder** | | n (%) | n (%) | n (%) |
| | Methadone | 106 (64.2) | 52 (63.4) | 54 (65.1) |
| | Buprenorphine | 10 (6.1) | 6 (7.3) | 4 (4.8) |
| | No medication for opioid use disorder | 50 (30.3) | 24 (29.3) | 26 (31.3) |

respectively. There was no significant difference in the baseline number of days or events of using injection drugs between AC and UC groups. More than 70% of individuals were receiving MOUD.

An average of 53.5 participants were in non-attendance at each follow-up visit throughout the four timepoints. The highest number of non-attendance occurred at the 3-month follow-up with 59 participants absent across both groups. There were no significant difference in the number of non-attendance between the AC and UC groups at all four follow-up time points.

The AC-by-time effect on the number of injection days in the past 30 days had an OR of 0.78 (95% CI = 0.62–0.98), demonstrating significantly decreased odds of reporting higher rates of injection days in the AC arm compared to the UC arm for each one unit increase in point of time (Table 3). This indicates a higher likelihood of an individual from the AC group to move to a category of a lower range of injection days over time (Fig 2). Similar results were seen in the number of injection events with AC-by-time effect with OR of 0.70 (95% CI = 0.56–0.88), indicating a higher likelihood of an AC participant being in a category of lower injection event frequencies over time (Fig 3).

Usage of a syringe previously used by someone else (receptive sharing) was rare throughout the study, with the baseline average number of times of receptive sharing of syringes in the last 30 days being 2.2 and 0.5 in the AC and UC groups respectively. The AC group saw a greater decrease in using syringes previously used by someone else (decrease by 1.5) than the UC group (decrease by 0.4) (Fig 4), however, this interaction effect was without statistical significance (OR = 0.79, [95% CI, 0.42–1.47]) (Table 4). There was a greater reduction in the average

**Table 3. Injection behavioral outcomes.**

| Category by frequency (days) | 0 | | 1–8 | | 9–29 | | 30 | |
|---|---|---|---|---|---|---|---|---|
| | AC | UC | AC | UC | AC | UC | AC | UC |
| **No. of days of injection in the last 30 days (%)** | | | | | | | | |
| **AC-by-time effect OR = 0.78 [95% CI 0.62–0.98]** | | | | | | | | |
| Baseline | 13 (15.8) | 13 (15.6) | 17 (20.7) | 26 (31.3) | 19 (23.2) | 12 (14.5) | 33 (40.2) | 32 (38.5) |
| 3 Months | 22 (37.9) | 20 (41.7) | 15 (25.9) | 10 (20.8) | 4 (6.9) | 10 (20.8) | 17 (29.3) | 8 (16.7) |
| 6 Months | 33 (55.9) | 22 (44.0) | 10 (16.9) | 5 (10.0) | 3 (5.1) | 4 (8.0) | 13 (22.0) | 19 (38.0) |
| 9 Months | 26 (44.8) | 25 (46.3) | 13 (22.4) | 10 (18.5) | 9 (15.5) | 6 (11.1) | 10 (17.2) | 13 (24.1) |
| 12 Months | 37 (57.8) | 24 (43.6) | 12 (18.7) | 13 (23.6) | 7 (10.9) | 4 (7.3) | 8 (12.5) | 14 (25.4) |
| **Category by frequency (no. of times)** | **0** | | **1–29** | | **30–89** | | **90+** | |
| | AC | UC | AC | UC | AC | UC | AC | UC |
| **No. of injection events in the last 30 days (%)** | | | | | | | | |
| **AC-by-time effect OR = 0.70 [95% CI 0.56–0.88]** | | | | | | | | |
| Baseline | 13 (15. 8) | 13 (15.7) | 17 (20.7) | 31 (37.3) | 20 (24.4) | 18 (21.7) | 32 (39.0) | 21 (25.3) |
| 3 Months | 22 (37.9) | 20 (41.7) | 16 (27.6) | 11 (22.9) | 6 (10.3) | 9 (18.7) | 14 (24.1) | 8 (16.7) |
| 6 Months | 33 (55.9) | 22 (44.0) | 11 (18.6) | 6 (12.0) | 5 (8.5) | 12 (24.0) | 10 (16.9) | 10 (20.0) |
| 9 Months | 26 (44.8) | 26 (48.1) | 18 (31.0) | 10 (18.5) | 8 (13.8) | 9 (16.7) | 6 (10.3) | 9 (16.7) |
| 12 Months | 37 (57.8) | 24 (43.6) | 17 (26.6) | 15 (27.3) | 4 (6.5) | 7 (12.7) | 6 (9.4) | 9 (16.4) |
| **No. receptive syringe sharing in the last 30 days (%)** | | | | | | | | |
| **AC-by-time effect OR = 0.79, [95% CI 0.42–1.47]** | | | | | | | | |
| Baseline | 69 (85.2) | 77 (92.8) | 9 (11.1) | 6 (7.2) | 3 (3.7) | 0 (0.0) | 0 (0.0) | 0 (0.0) |
| 3 Months | 56 (96.6) | 44 (91.7) | 2 (3.4) | 4 (8.3) | 0 (0.0) | 0 (0.0) | 0 (0.0) | 0 (0.0) |
| 6 Months | 58 (98.3) | 49 (98.0) | 1 (1.7) | 1 (2.0) | 0 (0.0) | 0 (0.0) | 0 (0.0) | 0 (0.0) |
| 9 Months | 58 (100.0) | 54 (100.0) | 0 (0.0) | 0 (0.0) | 0 (0.0) | 0 (0.0) | 0 (0.0) | 0(0.0) |
| 12 Months | 67 (95.7) | 58 (96.7) | 2 (2.9) | 2 (3.3) | 1 (1.4) | 0 (0.0) | 0 (0.0) | 0 (0.0) |
| **No. of times sharing cookers in the last 30 days (%)** | | | | | | | | |
| **AC-by-time effect OR = 0.68, [95% CI 0.47–0.97]** | | | | | | | | |
| Baseline | 62 (76.5) | 61 (73.5) | 13 (16.0) | 12 (14.5) | 5 (6.2) | 5 (6.0) | 1 (1.2) | 5(6.0) |
| 3 Months | 52 (89.7) | 39 (81.3) | 3 (5.2) | 5 (10.4) | 1 (1.7) | 3 (6.3) | 2 (3.4) | 1(2.1) |
| 6 Months | 52 (88.1) | 38 (76.0) | 5 (8.5) | 10 (20.0) | 0 (0.0) | 1 (2.0) | 2 (3.4) | 1(2.0) |
| 9 Months | 53 (91.4) | 45 (83.3) | 4 (6.9) | 9 (16.7) | 1 (1.7) | 0 (0.0) | 0 (0.0) | 0(0.0) |
| 12 Months | 66 (94.3) | 47 (78.3) | 2 (2.9) | 9 (15.0) | 1 (1.4) | 3 (5.0) | 1 (1.4) | 1(1.7) |

*Effect size = the effect of AC care over 12 months on the dependent variable

number of times cookers were shared in the last 30 days within the UC arm (decrease by 8.4) relative to the AC arm (decrease by 2.1) after 12 months (Fig 5). This difference in reduction was statistically significant (OR = 0.68, [95% CI 0.47–0.97]). The percentage of participants receiving MOUD decreased over the 12 month time period in both intervention arms (decrease by 18.8% and 20.4% in the AC and UC arms respectively). However, the differences in these decreases across time were not of statistical significance (OR = 0.83, [95% CI, 0.60–1.15], p-value = 0.263) (Fig 6). There was no isolated intervention-effect on injection frequency or behaviors.

Logistic regression was used to analyze the missingness of data at 12 months based on the five outcome variables during the previous four time points. The analysis revealed no significant association between the outcome variables and the missing data at 12 months, demonstrating that the outcomes were independent of the missing data. We also conducted a sensitivity analysis via multiple imputation. The only time effect or treatment-by-time

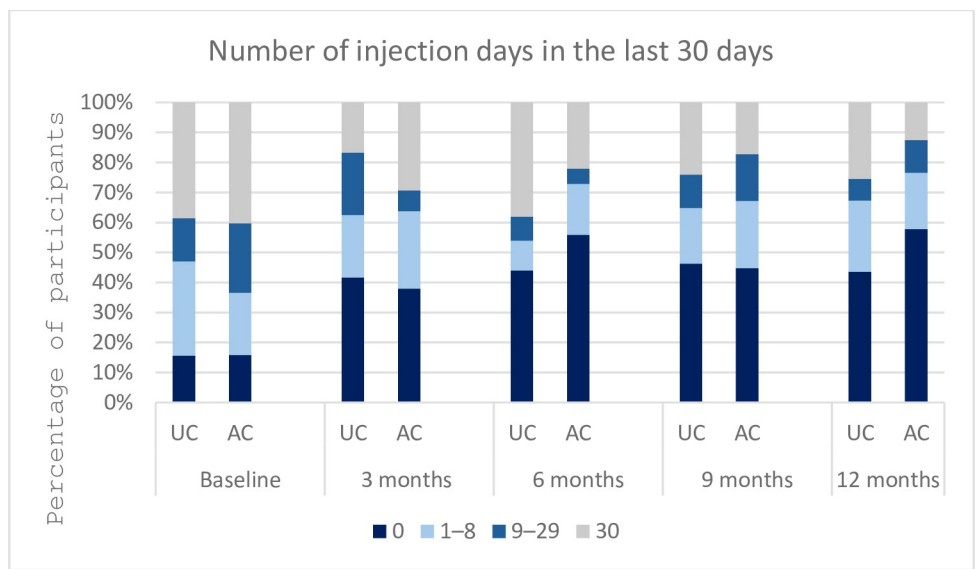

**Fig 2. Number of days with injection drug use in the last 30 days over 12 months.**

interaction effect that differed in terms of its significance was the treatment-by-time cooker sharing interaction effect, which became nonsignificant when multiple imputation was applied. Therefore, the results related to cooker sharing should be interpreted with caution.

## Discussion

### Frequency of injection drugs

In the current study, patients in the AC arm reported a significantly greater decrease in the total number of injection events and the number of days using IV substances within the last 30

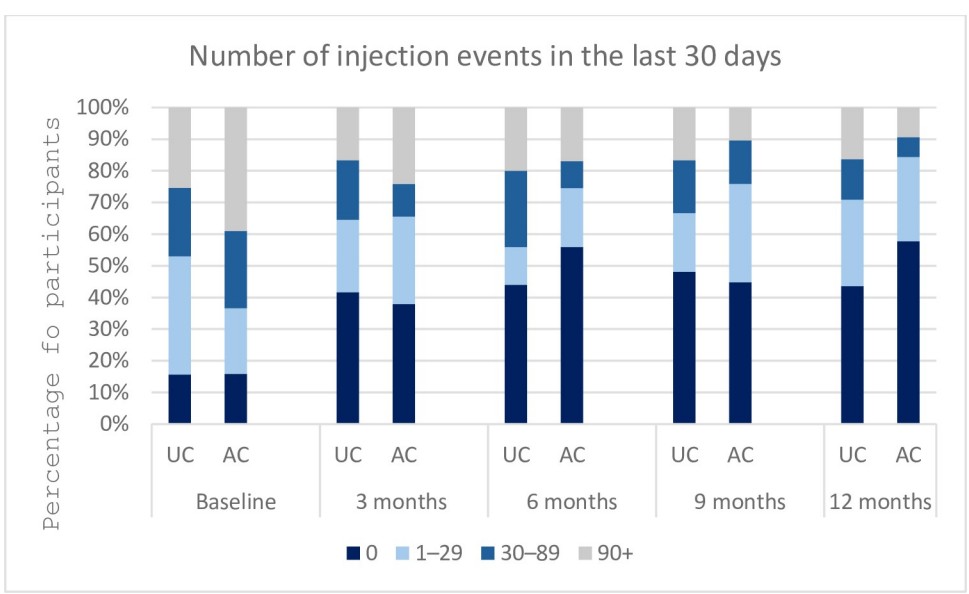

**Fig 3. Number of injection events in the last 30 days over 12 months.**

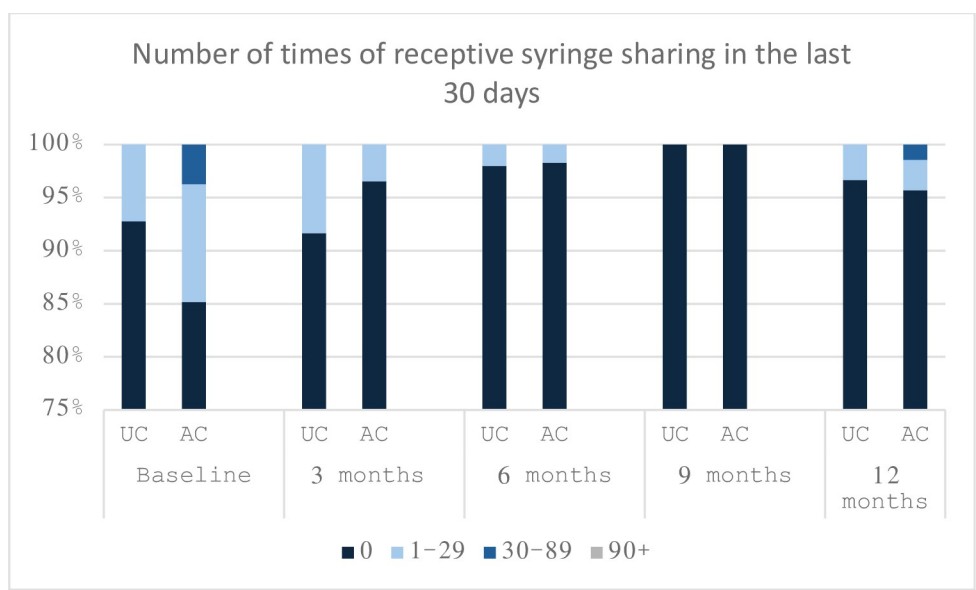

**Fig 4. Number of times using a syringe used by someone else in the last 30 days over 12 months.**

days throughout the 12 months of enrolment. While previous studies have looked at the independent impacts of HCV treatment and patient education on high-risk injection behaviors, our study is the first to analyze the collective impact of accessible HCV treatment, psychosocial support, and patient education for harm reduction delivered in a multi-disciplinary setting [16, 23, 24].

Our parent study showed that the higher rate of SVR12 achieved in the AC arm was not due to differences in treatment response but from higher rates of advancement along the care cascade prior to treatment initiation [9]. This is likely attributable to decreased psychosocial and logistical barriers to care [9]. It unfortunately remains commonplace for healthcare providers to attribute negative qualities such as "weak mindedness" to PWID [11, 20]. Experiences of such external stigma often translate into internal stigma and individuals begin to apply these negative qualities to themselves [11, 20]. In a 2020 survey, 78.1% of PWID reported stigmatizing experiences with healthcare providers (HCPs), highlighting the magnitude of the issue of demoralization that PWID experience secondary to stigmatization [25]. The emphasis that the AC arm placed on interactions devoid of stigmatization and paternalism likely increased the self-efficacy of participants, encouraging them to seek out medical care and make health-conscious decisions which resulted in the increased advancement through the care cascade and the decrease in injection frequency. Qualitative studies of HCV-treated

**Table 4. Changes in treatment for opioid use disorder.**

| Number of participants in treatment for opioid use disorder in the last 90 days (%) | | |
|---|---|---|
| AC-by-time effect OR = 0.83 [95% CI 0.60–1.15] | | |
| | AC | UC |
| Baseline | 67 (81.7%) | 64 (77.1%) |
| 3 Months | 38 (65.5%) | 36 (75.0%) |
| 6 Months | 38 (64.4%) | 25 (50.0%) |
| 9 Months | 41 (70.7%) | 31 (57.4%) |
| 12 Months | 44 (62.9%) | 34 (56.7%) |

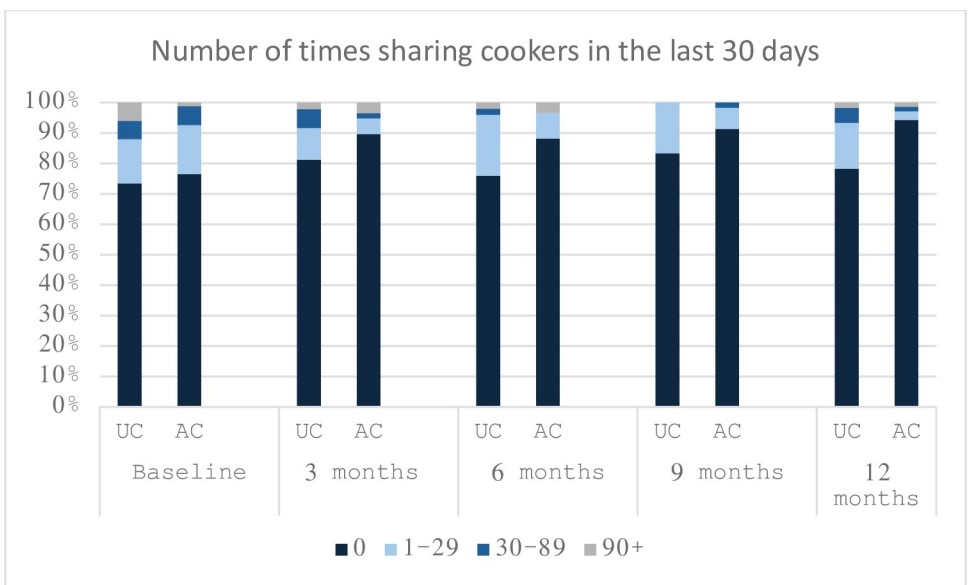

**Fig 5. Number of times sharing cookers in the last 30 days over 12 months.**

PWID suggest that receiving treatment on its own increases one's sense of responsibility over their wellbeing [20, 26]. The knowledge of having made a health-conscious decision paired with the measurable outcome of a cleared viral load can act as a motivating point of "transformation" for further behavioral changes including injection frequency and method of injection [20, 26]. Thus, the higher percentage of patients initiating treatment and achieving SVR12 in the AC arm could be a contributing factor to the greater decrease in injection frequency seen in patients in the AC arm.

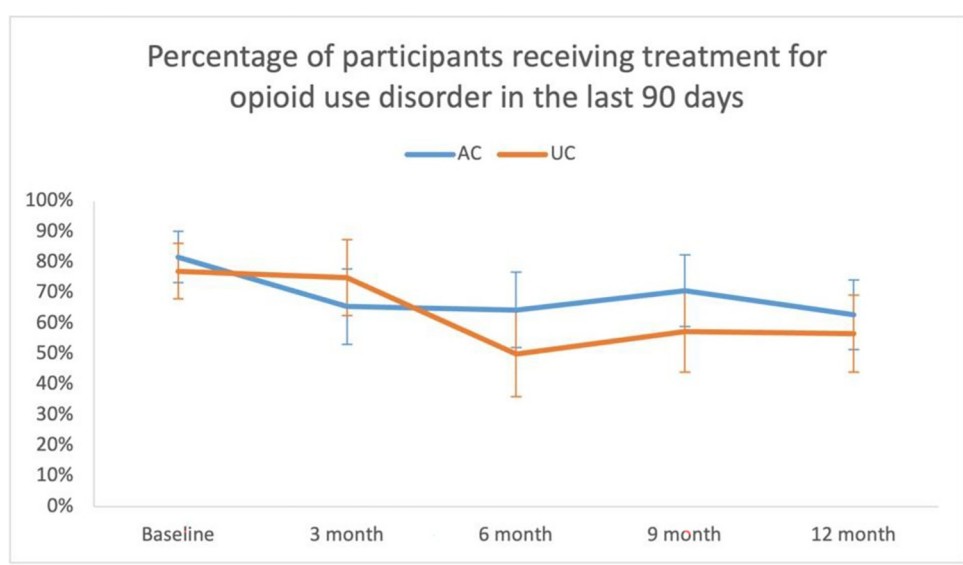

Bars represent 95% confidence intervals

**Fig 6. Percentage of participants receiving MOUD in the last 90 days over 12 months.**

## High-risk injection behaviors

There was no statistical difference in the rate of receptive sharing of syringes over the 12 months between those receiving AC and UC care. There was a decrease in the rate of cooker sharing in the AC group compared to the UC group. The statistical significance of this depended on whether missing data was considered or not. One possibility is that objects used as cookers are more readily available than syringes and so there was less of a barrier for participants to apply safe injection practices they had been taught against sharing cookers in the AC arm. Another possibility is that there was no difference between both syringe or cooker sharing rates between the AC and UC groups as implied when missing data is factored in. The decision to inject may be the rate-limiting step in the process of making a health-conscious decision, and that once the decision to inject has been made, the threshold for overlooking the health-risks has already been crossed.

A study led by Garfein et al. in 2002 compared the impact of peer education intervention (PEI) with video discussion intervention (VDI) on high-risk injection behaviors [23]. PEI comprised of 6 two-hour group sessions where PWID received education on harm reduction, community-based resources, and peer-education from an in-person facilitator [23]. Those in the VDI group were asked to watch videos addressing psychosocial issues, with less emphasis on harm reduction, with the same number of cumulative hours of education as the PEI group [23]. This trial looked at rates of receptive sharing of used syringes, use of a new syringe to divide drugs, sharing of cookers/cotton/rinse water, and the proportion of partners that participants shared paraphernalia with [23]. Interestingly, there was only a significant change in decrease in the composite index of all six outcomes in the PEI group compared to the VDI group. There was no significant difference in the rates of the six outcomes when individually compared between the two groups [23]. Contrastingly, Bertrand et al. conducted a study comparing individuals receiving motivational interviewing (MI) versus educational intervention (EI) [24]. At the 6 month mark, individuals in the MI group were 50% less likely to share paraphernalia excluding syringes (OR = 0.50; CI = 0.09–0.90, p = 0.017), and 53% less likely to share containers (OR = 0.47; CI = 0.11–0.84, p = 0.011) [24].

Data regarding change in high-risk injection behaviors is sparse. However, the data available, including our study and two studies mentioned above, is supportive of a trend towards decreased rates of high-risk injection behaviors when PWID receive interactive and personalized education on harm reduction and safe injection practices [28].

## Enrolment in MOUD programs

The rate of enrolment in MOUD programs fell in both arms throughout the 12 months with no statistical difference between the AC and UC arms. The magnitude of the decreases in both arms is consistent with the MOUD retention rates seen in prior studies [17, 27, 28]. Given our low-threshold care model did not require (or even endorse) abstinence or engagement in MOUD as a prerequisite to HCV treatment, it was not surprising to see no statistical difference between AC and UC in MOUD retention.

As our study took place amidst the pandemic, the impact of social distancing, given pre-COVID policies requiring in-person appointments for MOUD administration, was considered as a possible confounder. A sub-study of the parent study looked at the impact of the COVID-19 pandemic on drug service utilization by HCV-positive PWID in New York City [29]. Subjects were categorized into pre-COVID versus post-COVID groups as well as AC versus UC groups. There was no significant decrease in MOUD enrolment post-COVID except for a significant decrease in buprenorphine-based program enrolment post-pandemic in the UC

group [29]. However, the majority of the participants receiving MOUD in the current study were in methadone-based programs.

## Limitations

One of the limitations of our study is the non-attendance rates at the different follow-up time points during which interviews were held. This could be partially attributed to the fact that the study took place during the COVID-19 pandemic. The greatest rate of non-attendance occurred at three months with thirty-five and twenty-four participants from the UC and AC group respectively not attending the follow-up interviews. However, many participants who did not attend a follow-up visit attended subsequent visits and participated in questionnaires then. At the 12-month mark, there was no significant difference in the number of those lost to follow-up in the AC and UC cohorts.

Another limitation that we faced were the outliers when it came to injection frequency. To circumvent this issue, we presented and analyzed the data as ordinal categories that represented the distribution of frequencies rather than with mean values. This unfortunately could not be done due to the dichotomous distribution of answers we got regarding injection behaviors. Thus, the mean values presented for frequency of syringe and cooker sharing should be interpreted with caution as they account for outliers on both ends of the spectrum.

Given that the frequency of injection and high-risk behavior frequencies were self-reported during interviews, the possibility of recall bias and social desirability bias is present. In this study, questionnaires were administered by interviewers. Assuming that individuals in the AC arm had stronger rapport with the care team, it is important to consider whether this led to underreporting due to concerns of disappointing the interviewer or if it led to more transparency.

## Conclusion

Historical reluctance to treat HCV infections in PWID due to concerns of re-infection are now combatted with guidelines recommending prompt treatment regardless of IV substance use [30]. These recommendations are supported by recent studies showing low rates of re-infection and comparable rates of successful treatment in PWID compared to those who do not inject drugs [31–34]. As successful as the new data has been in encouraging HCPs to offer HCV screening and treatment, re-infection rates are not zero and PWID remain the highest-risk group for infection in developed countries [35]. Our parent study and the current secondary analysis elucidate the need for accessible multi-disciplinary HCV care shown to improve clinical outcomes and decrease frequency of injection drug use–promoting harm reduction [9]. However, even with the exponentially more comprehensive supports that the AC intervention offered relative to the current landscape of HCV care, there was a lack of significant impact on high-risk injection behaviors and MOUD enrolment which are crucial for ensuring long-term harm reduction. Further studies exploring AC with greater individualized psychosocial supports are needed for the medical community to appreciate the need for a biopsychosocial approach to decreasing HCV transmission amongst PWID–an essential step towards achieving the goal of HCV elimination.

## Supporting information

**S1 Checklist. Consort checklist.**
(DOC)

**S2 Checklist. Human subjects checklist.**
(DOCX)

**S1 File. Study protocol.**
(PDF)

**S2 File. SAS data file.**
(HTM)

## Author Contributions

**Conceptualization:** Pedro Mateu-Gelabert, Shashi N. Kapadia, Kristen M. Marks, Benjamin Eckhardt.

**Data curation:** Yesenia Aponte Melendez, Chunki Fong, Melinda Smith.

**Formal analysis:** Claire So Jeong Lee, Pedro Mateu-Gelabert, Yesenia Aponte Melendez, Chunki Fong, Shashi N. Kapadia, Kristen M. Marks, Benjamin Eckhardt.

**Investigation:** Pedro Mateu-Gelabert, Yesenia Aponte Melendez, Melinda Smith, Kristen M. Marks, Benjamin Eckhardt.

**Methodology:** Pedro Mateu-Gelabert, Yesenia Aponte Melendez, Shashi N. Kapadia, Melinda Smith, Kristen M. Marks, Benjamin Eckhardt.

**Project administration:** Pedro Mateu-Gelabert, Yesenia Aponte Melendez, Kristen M. Marks, Benjamin Eckhardt.

**Resources:** Benjamin Eckhardt.

**Software:** Chunki Fong.

**Supervision:** Pedro Mateu-Gelabert, Yesenia Aponte Melendez, Kristen M. Marks, Benjamin Eckhardt.

**Validation:** Pedro Mateu-Gelabert, Shashi N. Kapadia, Benjamin Eckhardt.

**Visualization:** Chunki Fong.

**Writing – original draft:** Claire So Jeong Lee, Pedro Mateu-Gelabert, Shashi N. Kapadia, Benjamin Eckhardt.

**Writing – review & editing:** Claire So Jeong Lee, Pedro Mateu-Gelabert, Yesenia Aponte Melendez, Chunki Fong, Shashi N. Kapadia, Kristen M. Marks, Benjamin Eckhardt.

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
