## [Decision Letter · Decision Letter 0]

1 May 2024

PONE-D-24-06895Reduced injection risk behavior with co-located Hepatitis C treatment at a syringe service program: Findings from the Accessible Care Randomized TrialPLOS ONE

Dear Dr. Eckhardt,

Thank you for submitting your manuscript to PLOS ONE. After careful consideration, we feel that it has merit but does not fully meet PLOS ONE’s publication criteria as it currently stands. Therefore, we invite you to submit a revised version of the manuscript that addresses the points raised during the review process.

We look forward to receiving your revised manuscript.

Kind regards,

Jason T. Blackard, PhD

Academic Editor

PLOS ONE

Journal Requirements:

PMG & KM R01 DA041298. National Institute on Drug Abuse. https://nida.nih.gov/.  No

SK K01 DA048172. National Institute on Drug Abuse. https://nida.nih.gov/.  No

KM  IN-US-342-4456. Gilead Sciences.  https://www.gilead.com/. No

I have read the journal's policy and the authors of this manuscript have the following competing interests: B. E., S. N. K., and K. M. M. have received research grants to their respective institutions from Gilead Sciences Inc.

We note that one or more of the authors are employed by a commercial company: Gilead Sciences Inc.

“The funder provided support in the form of salaries for authors, but did not have any additional role in the study design, data collection and analysis, decision to publish, or preparation of the manuscript. The specific roles of these authors are articulated in the ‘author contributions’ section.”

4. In the online submission form, you indicated that your data is available only on request from a third party. Please note that your Data Availability Statement is currently missing [the name of the third party contact or institution / contact details for the third party, such as an email address or a link to where data requests can be made]. Please update your statement with the missing information. 

Additional Editor Comments:

**This is a randomized control trial of HCV syringe services conducted in New York.**

**The purpose of this study is quite interesting.  However, the methods are a bit confusing and the results are not very surprising or compelling.**

**The terms “accessible” care versus “usual” care are not described in the abstract and require additional explanation for the casual reader.  Similarly, the explanation in the methods requires additional details as given in Supplementary Figure 1.**

**The authors should explain lines 145-146 in more detail:  The on-site treatment team facilitated a non-stigmatizing atmosphere during their interactions with the AC arm participants.**

**Lines 149-151:  is this information standardized in written / oral / virtual form?**

**The HCV care coordinator provided on-site education on HCV and prevention of re-infection, and support regarding treatment navigation, insurance, and adherence. **

**It is unclear what drugs were injected and if that had any impact on injection behaviors, SVR, and/or study visits.**

**The data presented in Supplementary Figure 1 should be explained in more detail in the first paragraph of the Results.  More context about the cohort is needed.**

Reviewers' comments:

Reviewer's Responses to Questions

**Comments to the Author**

1. Is the manuscript technically sound, and do the data support the conclusions?

Reviewer #1: Partly

Reviewer #2: No

Reviewer #3: Partly

2. Has the statistical analysis been performed appropriately and rigorously? 

Reviewer #1: No

Reviewer #2: No

Reviewer #3: No

3. Have the authors made all data underlying the findings in their manuscript fully available?

Reviewer #1: No

Reviewer #2: Yes

Reviewer #3: No

4. Is the manuscript presented in an intelligible fashion and written in standard English?

Reviewer #1: No

Reviewer #2: No

Reviewer #3: Yes

5. Review Comments to the Author

**Reviewer #1:** Here is a list of specific comments. Note: line and page numbering in reviews and comments is based on ruler applied in Editorial Manager-generated PDF.

1. Page 3, lines 48–51: I suggest considering reporting coefficients of interactions in a clinical term. In addition, I suggest reporting 95% confidence intervals.

2. Page 9, lines 190–191: I suggest adding ‘mean with standard deviation’ per tables.

3. Page 9, lines 191–192: These tests were not reported. I suggest excluding this sentence.

4. Page 9, lines 194–195: I suggest clarifying the calling out of Table 2. The title of Table 2 indicated “baseline.” However, the sentence described the ordinal category at all study points.

5. Page 9, lines 197–198: I recommend providing additional information about the generalized estimating equations (GEE) such as the distribution, the link function, and model specification. Also, please confirm the use of t-test in GEE. The most popular test in GEE is the Wald test with naive or robust standard errors.

6. Page 9, lines 198–200: I recommend assessing the balance in the numbers of days and events of injection drug use at baseline between arms and determining the need for adjustment based on this balance, rather than solely on statistical significance.

7. Page 10, Table 1: Were there any concerns regarding imbalance in the parent study?

8. Page 11, lines 220–221: Would it be possible to declare the balance without referring the statistical significance?

9. Page 12, lines 226–228: Can you please elaborate the average number of participants in non-attendance? It was not clear how to interpret the numbers of 53.5 and 59.

10. Page 12, lines 228–229: I suggest considering removing this sentence.

11. Page 12, line 231: Did the intervention-by-time effect analysis refer to the GEE with interaction terms between intervention and time? If so, I suggest providing the specifications in the Statistical Analysis section. See Comment #5 above.

12. Page 12, lines 233–239: Would you mind providing the SPSS outputs from GEE as a supplementary material or an attachment to the response? It was not easy to envision the models.

13. Page 13, Table 3: The coefficient of the interaction would only be directly interpretable if the outcome was continuous; i.e., the distribution of GEE was Gaussian and the link function was identity. Modeling the percentage as a continuous outcome might not be ideal but it was acceptable given GEE.

**Reviewer #2:** This paper describes a secondary analysis of a previously published paper and rehashes those findings at length rather than referring the reader to the publication and summarizing the results here in brief. Moreover, in the Discussion, the authors describe the key outcome differences in the parent study due to "...rates of advancement along the care cascade" (p. 18/lines 275-78), which is not mentioned in the previous description of the primary study.

There is no mention of possible recall bias or acceptability bias despite the fact that participants were asked to enumerate the frequency of specific drug use behaviors over a 30-day period.

In addition, there are a number of awkward phrasings (e.g. "within PWID" rather than "among PWID") that could benefit from a less stilted style.

The tables are long (esp Table 3) and I wonder if there is a more visual way to display this information.

**Reviewer #3:** This paper reports on the outcomes of a randomized clinical trial of an accessible care approach to providing Hepatitis C treatment relative to treatment as usual condition on the primary outcomes of frequency of injection drug use and engagement in high-risk injection drug use engagement. Results indicate that those randomized to the accessible care condition had lower frequency of injection drug use and high-risk injection practices relative to the treatment as usual condition.

The following are suggestions to improve the manuscript.

-In the introduction (pg. 5): It would be helpful to clarify what patient navigation in the control arm entailed.

-The authors note in the introduction that past, previously reported results of this trial indicated the accessible care approach was associated with higher sustained virologic response relative to the usual care arm. The authors should make more explicit in the abstract that the primary outcome of the trial was not the primary outcome(s) being reported in this paper (i.e., injection drug use frequency, frequency of high-risk injection practices).

-In the methods, I would recommend describing the extent to which content focused on altering injection practices was included in either the accessible care or usual care conditions. I would assume this counseling is part of all the SSP services provided, but wasn’t clear whether there was additional emphasis provided on reducing frequency of injection drug use in either condition.

-On page 9, the authors also indicate that uptake of MOUD is an outcome, but this hasn’t been addressed as an outcome in the manuscript to this point. There is a need for incorporating MOUD into the abstract and introduction.

-Additional description of the measurement of the manuscript’s primary outcomes is needed. Further, how were categories of number of days injecting determined? What was a continuous measure not employed? The rationale for separating 29 days from 30 days is unclear, for example.

-Additional information regarding the retention and attrition by condition is needed along with how missing data was handled in the analysis. Further description of modeling over all assessment points relative to examination at the 12-month follow-up is needed.

-Table 2: Not clear what first 2 rows represent. This should be revised.

-The presentation of results in Table 3 should be revised.

-Figures 3 and 4 should indicate what the bars represent. Reviewing the figures it would appear while there may be a statistically significant difference, there may not be clinically meaningful differences between the two conditions. Further, these figures are somewhat misleading if the analysis utilized the ordinal categories rather than the continuous measures.

-The authors list outliers as a limitation of this study. The rationale for ordinal categories is not well supported and it is unclear why the authors did not consider alternative analytic approaches.

-There were a few instances where tracked changes remained in the document that should be addressed. Figure 1 did not appear correctly in the final submitted document.

6. PLOS authors have the option to publish the peer review history of their article (what does this mean?). If published, this will include your full peer review and any attached files.

Reviewer #1: No

Reviewer #2: No

Reviewer #3: No

---

## [Author Response · Author response to Decision Letter 0]

20 Jun 2024

Thank you for reviewing our manuscript and for providing your feedback– helping make our manuscript stronger. Below are the reviewers comments (in bold) and our responses (in plain font). Edited text is shown in italic font. 

Reviewer 1

1. Page 3, lines 48–51: I suggest considering reporting coefficients of interactions in a clinical term. In addition, I suggest reporting 95% confidence intervals.

• We are now reporting our results in odds ratios instead of beta-coefficients. We hope that this makes our results easier to understand and apply clinically. We have described the interpretation of our results in the following sentences:

The AC-by-time effect on the number of injection days in the past 30 days had an OR of 0.78 (95% CI= 0.62-0.98), demonstrating significantly decreased odds of higher rates of injection days in the AC arm compared to the UC arm for each one unit increase in point of time (Table 3). This indicates a higher likelihood of an individual from the AC group to move to a category of a lower range of injection days over time (Figure 1). Similar results were seen in the number of injection events with AC-by-time effect with OR of 0.70 (95% CI=0.56-0.88), indicating a higher likelihood of an AC participant being in a category of lower injection event frequencies over time (Figure 2). (Line 269-276) 

• 95% confidence intervals have been added. 

2. Page 9, lines 190–191: I suggest adding ‘mean with standard deviation’ per tables.

Sentence has been reworded to include “mean with standard deviation”:

The baseline characteristic of participants were reported as frequencies with percentages and mean values with standard deviations. (Line 190-191) 

3. Page 9, lines 191–192: These tests were not reported. I suggest excluding this sentence.

Tests not reported/irrelevant have been excluded. 

4. Page 9, lines 194–195: I suggest clarifying the calling out of Table 2. The title of Table 2 indicated “baseline.” However, the sentence described the ordinal category at all study points.

Sentence has been changed to clarify the calling out of table 2: 

At baseline (Table 2) and at each study timepoint (Table 3), participants were grouped into ordinal categories of a range of number of injection events/days in the last 30 days based on their interview answers. (Line 199-201) 

5. Page 9, lines 197–198: I recommend providing additional information about the generalized estimating equations (GEE) such as the distribution, the link function, and model specification. Also, please confirm the use of t-test in GEE. The most popular test in GEE is the Wald test with naive or robust standard errors.

• The use of t-test has been confirmed (Line). 

• We changed our statistical analysis from GEE to generalized linear mixed model (GLMM) to incorporate random intercept effect to improve the model

• Details about the distribution, link function, and the models have been included:

-we modeled the data using generalized linear mixed model (GLMM) to evaluate the main effect of intervention, time, and the interaction effect of intervention and time. Additionally, initiation to treatment and its interaction with intervention were included as covariates, accounting for potential confounding effect among those who initiated treatment vs those who did not. Given our focus on testing for trends, time was treated as a numeric variable. For the treatment of opioid use disorder, since the variable is dichotomous, a binary distribution with logit link function was used. For injection frequency and equipment sharing frequency variables, a multinomial distribution with cumulative logit link function was used. All models included a random intercept. (Line 211-219) 

6. Page 9, lines 198–200: I recommend assessing the balance in the numbers of days and events of injection drug use at baseline between arms and determining the need for adjustment based on this balance, rather than solely on statistical significance.

Baseline characteristics, including number of days and events of injection use, were balanced between the AC and UC arms with the exception of the percentage of participants recently incarcerated being higher in the UC arm. The randomized controlled trial design of this study also ensured balance through randomization when assigning participants to the two arms. 

7. Page 10, Table 1: Were there any concerns regarding imbalance in the parent study?

The only imbalance between the baseline characteristics of participants in the two arms was the percentage of recent incarceration being higher in the UC arm compared to the AC arm. We have added a footnote under Table 1 and a line in the results section highlighting this:

b Statistical difference between AC and UC with p<0.05 (Table 1)

The baseline characteristics of the two arms were balanced except for the percentage of participants in the UC arm incarcerated in the last 90 (12.0%) being significantly higher than in the AC arm (2.4%). (Line 246-248)

8. Page 11, lines 220–221: Would it be possible to declare the balance without referring the statistical significance?

Given that this study was a randomized controlled trial the design itself ensured balance through randomization. We referred to the lack of statistical significance to further emphasize this and also provide the raw data in table 2. The following sentences have also been added to imply balance by inspection:

The majority of participants in both treatment arms were male (77.6%), with a mean age of 42.0. The AC group had more participants aged 45 or older than the UC group (45.1% vs. 36.1), while the UC group had more participants aged 31-44 (51.8% vs. 41.5%). There were slightly more Hispanic participants in the UC arm (62.7%) than in the AC arm (54.9%). The UC arm had 51.8% of participants report the use of heroin in the last 90 days compared to the 43.9% in the AC arm. In both groups, more than half of study participants experienced homelessness in the last 3 months. Most participants (84%) had a history of incarceration with 7.3% having been incarcerated within the last 90 days. Most participants (93.9%) were publicly insured. More participants in the AC arm had seen a non-HCV clinician in the last 90 days (53.7%) than participants in the UC arm (45.8%). The baseline characteristics of the two arms were balanced except for the percentage of participants in the UC arm incarcerated in the last 90 (12.0%) being significantly higher than in the AC arm (2.4%). (Line 236-248)

9. Page 12, lines 226–228: Can you please elaborate the average number of participants in non-attendance? It was not clear how to interpret the numbers of 53.5 and 59.

If the number of non-attendances between the two arms across all 4 time points were taken and averaged, the resulting number is 53.5. The highest number of non-attendances (59 non-attendances) occurred at the 3 month mark. The previous sentence has been worded: 

An average of 53.5 participants were in non-attendance at each follow-up visit throughout the four timepoints. The highest number of non-attendance occurred at the 3-month follow-up with 59 participants absent across both groups. (Line 264-266)

10. Page 12, lines 228–229: I suggest considering removing this sentence.

Given the theoretical possibility that there may be higher attendance rates at follow-up interviews in the AC group, we felt that it was important to explicitly state that there was no significant difference in the non-attendance rates between the AC and UC arms. 

11. Page 12, line 231: Did the intervention-by-time effect analysis refer to the GEE with interaction terms between intervention and time? If so, I suggest providing the specifications in the Statistical Analysis section. See Comment #5 above.

We are now using GLMM instead of GEE. Specifications have been provided in the statistical analysis section. Please see our response to comment #5 above.

12. Page 12, lines 233–239: Would you mind providing the SPSS outputs from GEE as a supplementary material or an attachment to the response? It was not easy to envision the models.

We have changed our model to the GLMM model. We have provided the SAS output as a supplementary material. 

13. Page 13, Table 3: The coefficient of the interaction would only be directly interpretable if the outcome was continuous; i.e., the distribution of GEE was Gaussian and the link function was identity. Modeling the percentage as a continuous outcome might not be ideal but it was acceptable given GEE.

We have changed the manuscript to report odds ratios using GLMM instead of beta-coefficient values. Please see our response to comment #5 above.

Reviewer 2

1. This paper describes a secondary analysis of a previously published paper and rehashes those findings at length rather than referring the reader to the publication and summarizing the results here in brief. 

We shortened the description and removed the results of the primary outcomes of the parent study in response to this comment. The retained details of the parent study were felt to be relevant to the outcomes of the current secondary analysis.

Deleted:

• The AC model provides low-threshold HCV treatment co-located at a syringe service program (SSP); involving proactive engagement with patients, flexible appointments, assistance in obtaining coverage for the cost of DAA agents, and a non-judgmental atmosphere which mitigates previously stigmatizing encounters PWID have experienced in healthcare settings.19

• The primary outcome of the parent Accessible Care study was the percentage participants achieving SVR12 within 12 months of study enrollment, which demonstrated a significantly higher SVR12 rate in the AC versus UC arm (67.1% vs. 22.9%, p<0.001).9

2. Moreover, in the Discussion, the authors describe the key outcome differences in the parent study due to "...rates of advancement along the care cascade" (p. 18/lines 275-78), which is not mentioned in the previous description of the primary study.

We have reworded the sentence to include this in the introduction where we are outlining the findings of the parent study and its relevance to the current secondary analysis:

The AC arm demonstrated significantly higher sustained virologic response (SVR) rates compared to the UC arm from higher rates of advancement along the care cascade in the AC arm . Successful treatment itself, in addition to a supportive care model, can be motivating for healthier lifestyles and choices.(9,20) (Line 110-113)

3. There is no mention of possible recall bias or acceptability bias despite the fact that participants were asked to enumerate the frequency of specific drug use behaviors over a 30-day period.

We have included the following in the limitations section:

Given that the frequency of injection and high-risk behavior frequencies were self-reported during interviews, the possibility of recall bias and social desirability bias is present. In this study, questionnaires were administered by interviewers. Assuming that individuals in the AC arm had stronger rapport with the care team, it is important to consider whether this led to underreporting due to concerns of disappointing the interviewer or if it led to more transparency. (Lines 409-414)

4. In addition, there are a number of awkward phrasings (e.g. "within PWID" rather than "among PWID") that could benefit from a less stilted style.

We have gone through the manuscript and have reworded awkwardly phrased sentences. 

5. The tables are long (esp Table 3) and I wonder if there is a more visual way to display this information.

Table 3 has been edited to be more succinct, have less white space, and present the data more clearly. 

Reviewer #3

This paper reports on the outcomes of a randomized clinical trial of an accessible care approach to providing Hepatitis C treatment relative to treatment as usual condition on the primary outcomes of frequency of injection drug use and engagement in high-risk injection drug use engagement. Results indicate that those randomized to the accessible care condition had lower frequency of injection drug use and high-risk injection practices relative to the treatment as usual condition.

The following are suggestions to improve the manuscript.

1. In the introduction (pg. 5): It would be helpful to clarify what patient navigation in the control arm entailed.

We reworded the introduction to clarify the difference between the AC arm and the UC arm (primarily referral based and much more heavily relied on patient navigation):

We previously reported results of a randomized controlled trial of the Accessible Care (AC) model which compared a supportive and on-site low-threshold treatment model (AC) to a referral and patient navigation-based model (usual care). (9,18) (Line 107-109)

We also revised the description of the UC arm in the methods section to further highlight the difference between the two arms:

Participants in the UC arm were connected to an on-site HCV patient navigator who was funded through the New York City Department of Health and part of the Check Hep C program. The patient navigator referred participants to HCV specialists at external hospitals/clinics for treatment and harm reduction programs. Similar to the AC arm’s care coordinator, the patient navigator also provided social support and assistance with insurance navigation. Contrastingly, support for treatment initiation and completion was not provided to the UC group. (Line 158-163)

2. The authors note in the introduction that past, previously reported results of this trial indicated the accessible care approach was associated with higher sustained virologic response relative to the usual care arm. The authors should make more explicit in the abstract that the primary outcome of the trial was not the primary outcome(s) being reported in this paper (i.e., injection drug use frequency, frequency of high-risk injection practices).

We have revised the last paragraph of the Introduction to make more explicit the primary outcome of the current secondary analysis:

To assess the changes in injection behaviors and practices of PWID associated with HCV treatment we analyzed data from the Accessible Care study.(6) The focus of this paper is to assess and compare the changes in injection behaviors, including injection frequency and frequency of high-risk injection behaviors, between PWID who received the AC intervention and usual care (UC) intervention. (Line 115-119) 

We have also edited the Methods section to re-emphasize the primary outcomes of this study:

The primary outcomes of this paper were changes in the frequency of high-risk injection behaviors (sharing of injection equipment and use of previously used syringes), injection frequency (number of days of injection and total number of injection events in the last 30 days), and uptake of MOUD at 3, 6, 9, and 12 months relative to baseline. (Line 181-185)

3. In the methods, I would recommend describing the extent to which content focused on altering injection practices was included in either the accessible care or usual care conditions. I would assume this counseling is part of all the SSP services provided, but wasn’t clear whether there was additional emphasis provided on reducing frequency of injection drug use in either condition.

Participants in the AC arm receiving DAA therapy were offered an on-site, interactive, oral presentation on re-infection prevention. Participants in the UC arm were not offered any on-site sessions. We have revised the methods section to make this difference clear:

Exclusive to the AC arm, all participants receiving DAA therapy were offered an on-site and interactive re-infection prevention oral presentation adapted from the validated Staying Safe Intervention program. (Line 153-155)

The patient navigator referred participants to HCV specialists at external hospitals/clinics for treatment. Similar to the AC arm’s care coordinator, the patient navigator also provided social support and assistance with insurance navigation. Contrastingly, support for treatment initiation and completion was not provided to the UC group. There was no on-site education regarding re-infection prevention offered. (Line 15

---

## [Decision Letter · Decision Letter 1]

17 Jul 2024

Reduced injection risk behavior with co-located Hepatitis C treatment at a syringe service program: The Accessible Care model

PONE-D-24-06895R1

Dear Dr. Eckhardt,

We’re pleased to inform you that your manuscript has been judged scientifically suitable for publication and will be formally accepted for publication once it meets all outstanding technical requirements.

Kind regards,

Jason T. Blackard, PhD

Academic Editor

PLOS ONE

Additional Editor Comments (optional):

None

Reviewers' comments:

Reviewer's Responses to Questions

**Comments to the Author**

1. If the authors have adequately addressed your comments raised in a previous round of review and you feel that this manuscript is now acceptable for publication, you may indicate that here to bypass the “Comments to the Author” section, enter your conflict of interest statement in the “Confidential to Editor” section, and submit your "Accept" recommendation.

Reviewer #1: All comments have been addressed

Reviewer #3: All comments have been addressed

2. Is the manuscript technically sound, and do the data support the conclusions?

Reviewer #1: Yes

Reviewer #3: Yes

3. Has the statistical analysis been performed appropriately and rigorously? 

Reviewer #1: Yes

Reviewer #3: Yes

4. Have the authors made all data underlying the findings in their manuscript fully available?

Reviewer #1: No

Reviewer #3: (No Response)

5. Is the manuscript presented in an intelligible fashion and written in standard English?

Reviewer #1: Yes

Reviewer #3: No

6. Review Comments to the Author

Reviewer #1: (No Response)

Reviewer #3: (No Response)

7. PLOS authors have the option to publish the peer review history of their article (what does this mean?). If published, this will include your full peer review and any attached files.

Reviewer #1: No

Reviewer #3: No

---

## [Editor Report · Acceptance letter]

20 Aug 2024

PONE-D-24-06895R1 

PLOS ONE

Dear Dr. Eckhardt, 

I'm pleased to inform you that your manuscript has been deemed suitable for publication in PLOS ONE. Congratulations! Your manuscript is now being handed over to our production team.

Kind regards, 

on behalf of

Dr. Jason T. Blackard 

Academic Editor

PLOS ONE